# The Role of a “Treat-to-Target” Approach in the Long-Term Renal Outcomes of Patients with Gout

**DOI:** 10.3390/jcm8071067

**Published:** 2019-07-20

**Authors:** Woo-Joong Kim, Jung Soo Song, Sang Tae Choi

**Affiliations:** Division of Rheumatology, Department of Internal Medicine, Chung-Ang University College of Medicine, Seoul 06974, Korea

**Keywords:** gout, urate-lowering therapy, renal function, chronic kidney disease

## Abstract

*Background*: Although gout is accompanied by the substantial burden of kidney disease, there are limited data to assess renal function as a therapeutic target. This study evaluated the importance of implementing a “treat-to-target” approach in relation to renal outcomes. *Methods*: Patients with gout who underwent continuous urate-lowering therapy (ULT) for at least 12 months were included. The effect of ULT on renal function was investigated by means of a sequential comparison of the estimated glomerular filtration rate (eGFR). *Results*: Improvement in renal function was only demonstrated in subjects in whom the serum urate target of <6 mg/dL was achieved (76.40 ± 18.81 mL/min/1.73 m^2^ vs. 80.30 ± 20.41 mL/min/1.73 m^2^, *p* < 0.001). A significant difference in the mean change in eGFR with respect to serum urate target achievement was shown in individuals with chronic kidney disease stage 3 (−0.35 ± 3.87 mL/min/1.73 m^2^ vs. 5.33 ± 11.64 mL/min/1.73 m^2^, *p* = 0.019). Multivariable analysis predicted that patients ≥65 years old had a decreased likelihood of improvement (OR 0.31, 95% CI 0.13–0.75, *p* = 0.009). *Conclusions*: The “treat-to-target” approach in the long-term management of gout is associated with better renal outcomes, with a greater impact on those with impaired renal function.

## 1. Introduction

Gout is one of the most common forms of inflammatory arthritis, which is caused by the deposition of monosodium urate crystals in and around the joints. There is an uneven distribution of gout worldwide; however, its incidence and prevalence continue to increase in many developed countries [1]. In patients with gout, comorbidities are common, with renal problems being top tiered. This generally leads to a substantial healthcare burden [2]. It is well known that the risk of gout appears to dramatically increase with decreasing glomerular function [3]. Approximately 25% of patients with gout evidenced chronic kidney disease (CKD) stage 3 or greater (defined as an estimated glomerular filtration rate (eGFR) of <60 mL/min/1.73 m^2^), with patients having gout being 78% more likely to develop CKD stage 3 or greater [4,5]. Moreover, gout is associated with a 57% increased hazard for the development of end-stage renal disease [6].

Hyperuricemia, a necessary predisposing factor for monosodium urate crystal deposition, can develop as a result of overproduction within the purine metabolism, or renal or gastrointestinal underexcretion. As uric acid is excreted primarily by the kidneys, renal underexcretion is considered to be responsible for the hyperuricemic mechanism in most patients with gout [7,8]. Sustained hyperuricemia facilitates kidney disease progression [9,10], and contributes to the pathogenesis of acute and chronic renal failure [11,12,13]. Serum urate concentration was proposed as an independent predictor for incident CKD [14], and a small number of controlled trials reported that lowering serum urate concentrations with xanthine oxidase inhibitors could help slow down CKD progression [15,16,17].

Over the last decade, the “treat-to-target” approach for the management of gout has been evaluated and recommended by rheumatology societies [18]. The current recommendations of the American College of Rheumatology [19] and the European League Against Rheumatism (EULAR) [20] both acknowledge that urate-lowering therapy (ULT) should generally be indicated for those with recurrent gout flares (>1 flare a year), tophi, CKD stage 2 or greater, or kidney stones. For individuals with gout who are starting ULT, the selection of a target serum urate concentration according to disease severity is important. Serum urate concentration should be monitored and maintained to <6 mg/dL, and therapeutic serum urate lowering to <5 mg/dL to facilitate faster dissolution of crystals is strongly recommended for patients with severe gout (tophi, chronic arthropathy, frequent gout flares) until total resolution. On the contrary, a clinical practice guideline published by the American College of Physicians (ACP) [21] recommends against this “treat-to-target” approach, noting that there is insufficient evidence to determine whether the benefits of escalating ULT to reach a target serum urate level outweigh the harms associated with repeated monitoring and increased medication.

Several efforts have been made to identify the relationship between renal function and ULT in patients with gout in previous research studies. A controlled study involving 59 participants, conducted more than three decades ago, reported that treatment with allopurinol did not change renal function but that retardation of the apparent decline was observed [22]. A post hoc analysis in the clinical trial of febuxostat suggested that individuals with greater reductions in serum urate level may experience stabilization of renal function decline [23]. However, the post hoc data of a randomized controlled trial of allopurinol showed no significant change in either serum creatinine or eGFR in patients with normal renal function [24]. The use of xanthine oxidase inhibitors was advocated in a small prospective study consisting of 106 primary gout patients and 51 healthy controls due to its beneficial effects on renal function [25]. Febuxostat proved to be effective in serum urate reduction in 96 patients with gout having moderate-to-severe renal impairment; however, there was no significant difference in renal function after one year [26]. A recent observation of a large prospective cohort including 4760 newly diagnosed patients with gout revealed that the initiation of allopurinol therapy of at least 300 mg/day was associated with a lower risk of renal function deterioration [27].

Acknowledging the scarcity of evidence to explicitly support the existence of a reno-protective effect from instituting ULT in patients with gout, there remains an area of uncertainty concerning the role of ULT in the development and progression of comorbid kidney disease, and it is still a matter of debate as to whether serum urate concentrations should be maintained below a certain target value. In the present study, we aimed to demonstrate the clinical importance of the “treat-to-target” approach in the long-term management of gout by evaluating the changes in renal function and the factors influencing the renal outcomes.

## 2. Methods

### 2.1. Patients

In this study, we identified individuals who were newly prescribed with urate-lowering agents, namely, allopurinol, febuxostat, or benzbromarone, after a working diagnosis of gout had been established by a board-certified rheumatologist in a single tertiary hospital between 1 March 2007, and 28 February 2017. Those who had not undergone prior treatment with urate-lowering agents and had at least 12 months of continuous ULT were included in the study. The exclusion criteria were: <20 years of age, pregnancy or lactation, a history of dialysis or renal transplantation, initial combination ULT, non-persistence (defined as the occurrence of the first gap in time covered by ULT prescription of at least 30 days after the end of the previous prescription), treatment for any malignant neoplasm during the study period, or no available clinical data.

### 2.2. Study Design

We reviewed the electronic medical record databases and retrieved the patients’ demographic characteristics and clinical data, including age; sex; body mass index (BMI); comorbidities; concomitant medication use; selected urate-lowering agent; adverse events during ULT; and laboratory-measured serum parameters such as serum urate concentration, serum creatinine concentration, eGFR, and liver function tests (LFTs). The index date was defined as the date of the first prescription of the ULT. Data available in the period from 30 days prior to the index date up until the index date were collected as baseline data. In accordance with the recommendations published by the EULAR [28], the dose of the selected urate-lowering agent had been gradually increased until the desired response was achieved, which was the maintenance of serum urate concentrations below the predefined target, if deemed clinically appropriate. The choice of whether to use nonsteroidal anti-inflammatory drugs, corticosteroids, or colchicine for the treatment of gout flares and for anti-inflammatory prophylaxis was left to the discretion of the treating physicians. All subjects were followed from the index date until (1) the time point of switching to or adding another urate-lowering agent, (2) the earliest episode of dialysis, (3) lost to follow-up, (4) death, or (5) 28 February 2018, whichever occurred first.

### 2.3. Outcomes

All subjects had at least one measurement of serum parameters within one month prior to and more than 12 months after the index date, in order to address the long-term effectiveness of ULT. The primary objective of this study was to evaluate the impact of serum urate reduction below target concentration of 6 mg/dL on renal function as measured by changes in eGFR values. The sequential comparisons of eGFR from baseline to the last follow-up were computed for the categorized eGFR at baseline. The dependent variable used in logistic regression analysis as a dichotomous binary variable was an increase in eGFR, which means that the last measured eGFR value is greater than the baseline value. The eGFR values were calculated using the abbreviated Modification of Diet in Renal Disease formula (eGFR = 175 × creatinine^−1.154^ × age^−0.203^ × 1.212 (if black) × 0.742 (if female)) [29], and the eGFR categories were classified according to the Kidney Disease: Improving Global Outcomes guideline [30]. The following categories were defined: normal (eGFR ≥ 90 mL/min/1.73 m^2^), mild renal impairment (eGFR 60–89 mL/min/1.73 m^2^), moderate renal impairment (eGFR 30–59 mL/min/1.73 m^2^), severe renal impairment (eGFR 15–29 mL/min/1.73 m^2^), and renal failure (eGFR < 15 mL/min/1.73 m^2^ or dialysis requirement). We defined acute kidney injury (AKI) as having a 1.5-fold increase in serum creatinine levels over baseline at any time during ULT. This is in accordance with the internationally accepted criteria [31], as well as newly developed urolithiasis when an agreement with the referral urologist has been reached.

### 2.4. Statistical Analysis

Descriptive analyses included absolute and relative frequencies for categorical variables as well as the mean ± standard deviation (SD) for approximately normally distributed quantitative variables, or median and interquartile range (IQR) for skewed quantitative variables. Between-group differences were assessed using the Pearson chi-square test and Fisher’s exact test for categorical variables, the independent *t*-test or the analysis of variance for normally distributed continuous variables, and the Mann–Whitney U test or Kruskal–Wallis test for non-normally distributed continuous variables. Within-group repeated-measure parametric data were compared using the paired *t*-test. The analysis of covariance model was constructed to assess the comparative effectiveness among the urate-lowering agents upon the increase in eGFR and included the prescribed agent as a factor with the baseline eGFR as a covariate. A multivariable logistic regression analysis was performed to evaluate the factors associated with the improvement in renal function, and a *p* value of 0.05 was used as a threshold for predictors to contribute significantly. Results of the logistic regression were presented as estimated odds ratio (OR) with 95% confidence interval (CI). All *p* values were two-sided, and *p*-values of less than 0.05 were considered statistically significant, unless otherwise specified. Data analyses were performed using IBM SPSS Statistics for Windows, Version 23.0. (IBM Corp., Armonk, NY, USA).

### 2.5. Ethical Considerations

All procedures involving human subjects were in accordance with the ethical standards of the institutional and national bioethics committee. The study protocol was reviewed and approved by the institutional review board of the Chung-Ang University Hospital, and informed consent was waived by the board because of the retrospective nature of our study (1810-012-16215).

## 3. Results

### 3.1. Subject Characteristics

We identified 555 patients with a diagnosis of gout and de novo ULT which continued for more than 12 months. After the exclusion of individuals who did not meet the enrolment criteria, 244 subjects were included for analysis (Figure 1). The study population was predominantly male (96.7%). The mean age was 50.9 ± 14.2 years, and 17.6% were ≥65 years old. BMI data were recorded in 202 subjects, with a mean value of 26.2 ± 3.1 kg/m^2^. The mean serum urate concentration was 7.9 ± 2.0 mg/dL, and the mean serum creatinine concentration was 1.08 ± 0.33 mg/dL. The mean eGFR among the study population was 77.12 ± 19.50 mL/min/1.73 m^2^, and 20.1% had an eGFR lower than 60 mL/min/1.73 m^2^. The median ULT duration was 25.3 months (12.2–126.4 months). Compared to patients who received one of the xanthine oxidase inhibitors, benzbromarone users had a lower baseline eGFR (*p* < 0.001) and a longer ULT duration (*p* < 0.001) (Table 1).

At the end of the follow-up periods, the mean serum urate concentration was significantly decreased (7.9 ± 2.0 mg/dL vs. 4.9 ± 1.6 mg/dL, *p* < 0.001). Of the 244 individuals, 191 (78.3%) achieved the target serum urate concentration of below 6.0 mg/dL at their last recorded data (52.4% for allopurinol users, 86.2% for febuxostat users, and 77.2% for benzbromarone users; *p* < 0.001). A total of 149 (61.1%) patients had a serum urate concentration below 5.0 mg/dL, 16 (6.6%) below 3.0 mg/dL, but none below 2.0 mg/dL.

### 3.2. Long-Term Renal Outcomes

While there was no improvement in renal function in patients who failed to attain the target serum urate concentration (79.72 ± 21.80 mL/min/1.73 m^2^ vs. 82.40 ± 23.21 mL/min/1.73 m^2^, *p* = 0.215), patients who achieved the target serum urate concentration had an improvement in renal function (76.40 ± 18.81 mL/min/1.73 m^2^ vs. 80.30 ± 20.41 mL/min/1.73 m^2^, *p* < 0.001), as depicted in Figure 2. Clinical characteristics were generally comparable between the two groups, except for the mean age at baseline, which was higher in patients who attained the serum urate target (46.3 ± 14.8 years vs. 52.2 ± 13.9 years, *p* = 0.008) (Table 2).

Table 3 presents the mean changes in eGFR according to the baseline eGFR categories. A statistically significant difference in the mean change in eGFR with respect to serum urate target achievement was only demonstrated in the subjects with moderate renal impairment, corresponding to CKD stage 3 (−0.35 ± 3.87 mL/min/1.73 m^2^ vs. 5.33 ± 11.64 mL/min/1.73 m^2^, *p* = 0.019). No significant difference was found between the two groups of demographic and clinical characteristics at baseline, with the use of angiotensin-converting enzyme inhibitors or angiotensin receptor blockers and diuretics (Appendix A). There was no significant difference in the mean change in eGFR for patients with better renal function; however, substantial increments in eGFR were observed irrespective of serum urate target achievement in patients with mild renal impairment.

It was noted that eight subjects who developed AKI during ULT had a considerable decline in the mean eGFR as opposed to the rest of the patients (−13.21 ± 16.26 mL/min/1.73 m^2^ vs. 4.23 ± 12.73 mL/min/1.73 m^2^, *p* < 0.001). There was no significant difference in the mean changes in eGFR with respect to the development of urolithiasis (*p* = 0.215). Despite the differences in the baseline eGFR values and the length of ULT duration, there was no significant difference in terms of the proportion of subjects with improved renal function (*p* = 0.543) and the mean change in eGFR (*p* = 0.101) among the three groups of urate-lowering agents selected for ULT.

### 3.3. Predictors of the Improvement in Renal Function

Improvement in renal function, as evidenced by an increase in eGFR from baseline, was reported in 147 (60.2%) patients during the study period. Logistic regression analysis was performed to evaluate the factors associated with the improvement in renal function following ULT as a function of baseline demographic and clinical variables (Table 4). A multivariable analysis identified that mild and moderate renal impairment were positively associated factors (OR 2.39, 95% CI 1.22–4.68, *p* = 0.012; OR 2.44, 95% CI 0.96–6.21, *p* = 0.060; respectively), while an age of ≥65 years was found to be a significant inverse predictor (OR 0.31, 95% CI 0.13–0.75, *p* = 0.009).

### 3.4. Adverse Events

There was no significant difference in terms of adverse event rates for each urate-lowering agent, and no serious adverse event leading to ULT discontinuation occurred during the study period (Appendix A).

## 4. Discussion

In this 10-year study evaluating the long-term renal outcomes of ULT in patients with gout, we obtained the following findings: (1) a significant increase in eGFR was only demonstrated in members of the study population that attained the target serum urate concentration; (2) a greater clinical benefit of the “treat-to-target” approach was observed in patients with renal impairment; (3) a decreased likelihood of improvement in renal function was expected in elderly patients.

The concept of ULT in gout was originally established to reduce the frequency of gout flares and prevent progressive joint destruction. The treatment goal for maintaining serum urate concentrations below 6.0 mg/dL, which is sufficiently below the crystal-forming saturation concentration of 6.8 mg/dL, was not specifically examined in a prospective manner. Instead, it has been employed following on from the cumulative evidence collected in pursuing complete suppression of flares and regression of tophi [32,33,34]. Although the concept of a “treat-to-target” approach rather than a “treat-to-symptoms” approach has been widely accepted by rheumatologists, it has not yet been validated to promote multiple health-related outcomes beyond the joints.

An annual eGFR decline of 2.5 mL/min/1.73 m^2^ is expected in untreated hyperuricemic adults, whereas the annual rate of decline in eGFR is 0.8 to 1.2 mL/min/1.73 m^2^ in healthy adults [35,36]. Perspectives on improving kidney disease through maintenance of serum urate below the target level were presented in recent observational studies. Statistically significantly lower likelihood of kidney disease progression resulted from long-term ULT, and individuals who had achieved the target serum urate concentration by dose adjustment had better renal outcomes compared to those maintained on the initial dosage, as reported by Kim et al. [37]. This scheme was supported by Levy and colleagues [38,39] who claimed that lowering the serum urate concentration below 6 mg/dL resulted in higher rates of eGFR improvement. However, these investigations have several notable limitations, including the fact that studied populations were composed of patients with gout and those without gout and the fact that only a few of them had a urate-lowering agent alternative to allopurinol, thus raising the question of whether serum urate reduction or merely the allopurinol itself led to the putative mechanism of benefits.

As broadly recognized, CKD stage 3 is associated with a greater medication burden, an increased incidence of renal failure, and poorer rates of survival [40,41]. Since the prevalence of gout among the population with CKD stage 3 has been reported to be 22.8%–25.5% [42,43], our results would be beneficial to many patients. We also note that a long-term intensive ULT have improved a larger portion of patients with mild renal impairment, regardless of their final serum urate concentrations. Overall, this study highlights the emerging insights into the renal outcomes in patients with gout, which capitalize on the “treat-to-target” approach already being recommended for joint protection. Accordingly, the ACP guidance emphasizing the value of minimizing symptoms and costs is not reflective of the long-term benefits of ULT and should be pursued only in exceptional circumstances.

There is a remarkable finding as regards the potential for kidney damage and ultimately worse outcomes following the development of AKI during ULT. Careful preventive measures are insufficient in themselves and monitoring of renal function is also warranted to detect AKI early, allowing for appropriate interventions to be implemented. It has been established that elderly patients are predicted to have a 69% lower chance of experiencing improvement in renal function. Given the moderate impairments of renal function prevalent among older adults [44], and the age-related polypharmacy often increasing the susceptibility to develop AKI, it is essential to preserve the renal function in this population. Clinicians should remain aware of the increased risks of this vulnerable population, which justifies the active surveillance for comorbid kidney disease.

Poor adherence and persistence are serious issues in real-world management of gout since only 47% of patients appeared to adhere to the prescription, and this has constantly proven to be a strong predictor of serum urate target achievement in prior contributions [45,46,47,48]. It was also indicated that 58% of patients were estimated to be non-persistent (a first medication gap of ≥30 days) at 1 year following the initiation of allopurinol therapy [49]. In this regard, this study was aimed to test hypotheses in such a setting as confounders can be minimized to the greatest degree as possible. In order to prevent yielding misleading outcomes on effectiveness of ULT, we reviewed the persistence of screened patients before embarking on this study to exclude poorly compliant subjects. However, due to this strict enrollment, care should be taken when generalizing the results to patients in routine clinical practice.

Another point of interest is that relatively fewer patients who received allopurinol as their urate-lowering agent reached the therapeutic target of serum urate below 6 mg/dL. While this poses the problem of inadequate serum urate reduction with allopurinol, it does not necessarily indicate that allopurinol itself is less suitable for gout management. There is a lack of an accepted best practice for allopurinol maintenance dosing, particularly in patients with CKD. As a result, allopurinol has been frequently underdosed in clinical practice, probably because a potentially lethal systemic hypersensitivity reaction is greatly feared, especially in East Asian societies [50,51]. Nevertheless, recent studies have revealed no association between the maintenance dose of allopurinol and the likelihood of allopurinol hypersensitivity syndrome development, and it has been suggested that a gradual dose escalation of allopurinol, even in patients with renal impairment, would be effective and well tolerated [52,53,54,55]. However, it remains possible that some subjects in this study may have received insufficient doses of allopurinol. It should also be noted that a relatively small number of patients were prescribed allopurinol in this study. This parallels the rapidly increasing market share of febuxostat within the recent prescription pattern in Korea [56].

On the other hand, benzbromarone is a potent uricosuric agent currently available outside the United States. The body of evidence supporting its effectiveness on renal function in patients with gout was limited by the paucity of prior studies and the concern that a uricosuric agent may not be effective for patients with CKD. However, it has been proposed in the retrospective cohort study that benzbromarone therapy could be more effective than allopurinol in reducing the risk of progression to dialysis among CKD patients [57].

Allopurinol, febuxostat, and benzbromarone were all regarded to be highly effective in achieving serum urate targets, as noted in a network meta-analysis by Li et al. [58]. However, there is limited data referencing kidney disease in patients with gout. Unfortunately, given the complexity of the analyses between heterogeneous groups, a definite conclusion in the comparative effectiveness of the urate-lowering agents for serum urate reduction or eGFR augmentation could not be drawn from this study. Future randomized controlled trials are required to confirm which agent is optimal for lifelong management in terms of better renal outcomes.

This study has several limitations emanating from its retrospective and uncontrolled nature. Recognizing the diagnostic uncertainty with reliance on administrative data means that features of gout were not always precisely recorded. Remarks about gout flares were not retrieved for analyses. Moreover, these are subject to being unreported or significantly underreported to the healthcare system. Some clinical data were missing, or incomplete, while non-severe adverse events might not be mentioned. However, the data we have analyzed are missing only BMI and the use of concomitant medications for few patients. Since trajectories of change in serum urate and eGFR over the course of treatment were not incorporated in this study, the pattern of improvement in eGFR after ULT had been commenced cannot be detected. Outcomes for the comparative group regarding the “treat-to-symptoms” approach or those groups with a fixed ULT dosage were not obtained in the study, while the effect of the “treat-to-target” approach with the upward titration of ULT dose has been established. We were not able to control for the concomitant use of either prescribed or over-the-counter medications that might act as potential confounders on the renal outcome.

We acknowledge that the sample size was not large, with a minimum follow-up duration of 12 months, which might have resulted in a lack of statistical power to identify all aspects of the reno-protective effect attributable to ULT. In addition, few subjects with severely impaired renal function, specifically CKD stage 4 and 5, were included. Given the lack of data, it was not possible to verify the renal benefit in those with already advanced kidney disease.

## 5. Conclusions

This study clarifies that long-term serum urate lowering to below 6 mg/dL after initiation of ULT is associated with the significant improvement of renal function in patients with gout, which has proven to be highly beneficial for conditions associated with renal impairment. The proper utilization of the “treat-to-target” approach in clinical practice would provide implications for better management and outcomes not only in gouty arthritis but also in comorbid kidney disease.

## Figures and Tables

**Figure 1 jcm-08-01067-f001:**
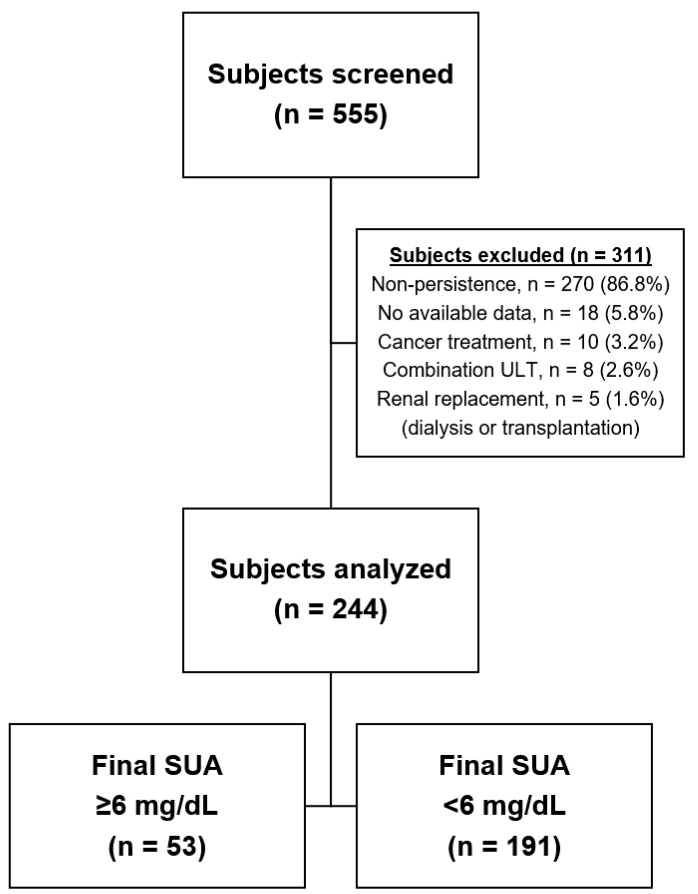
Study flow chart. Data represents enrolled population.

**Figure 2 jcm-08-01067-f002:**
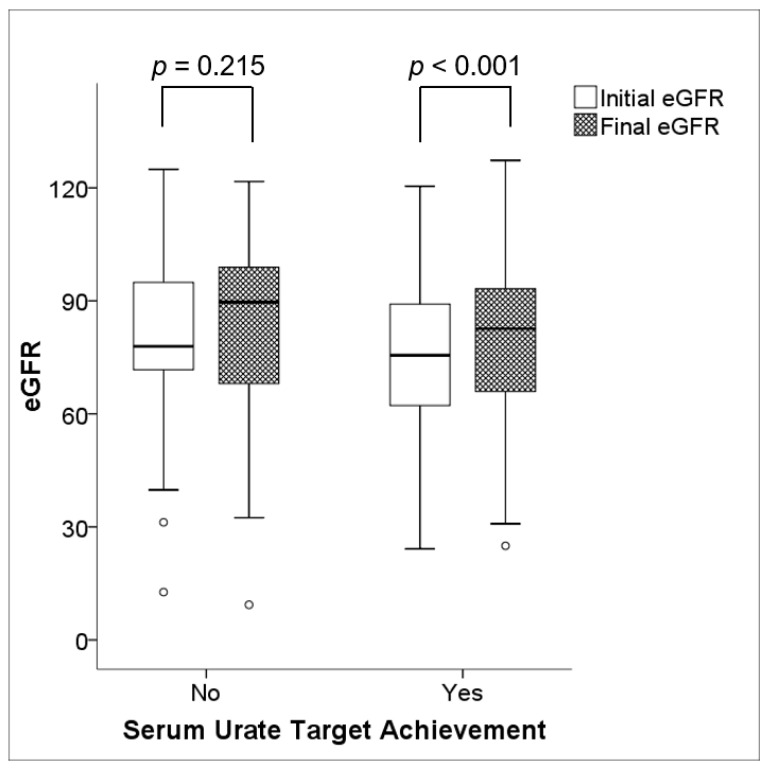
Changes in renal function during ULT. There was no significant change in renal function (*p* = 0.215) in the population without serum urate target achievement; however, a significant increment in renal function (*p* < 0.001) was demonstrated in the population with serum urate target achievement. Note: eGFR values are expressed in mL/min/1.73 m^2.^

**Table 1 jcm-08-01067-t001:** Baseline demographic and clinical characteristics of the study population.

Characteristics	Allopurinol (*n* = 42)	Febuxostat (*n* = 145)	Benzbromarone (*n* = 57)	Overall (*n* = 244)	*p* Value
Male sex, number (%)	39 (92.9)	140 (96.6)	57 (100.0)	236 (96.7)	0.113
Age, years	55.1 ± 15.8	49.5 ± 13.5	51.4 ± 14.5	50.9 ± 14.2	0.073
Age of ≥65 years, number (%)	12 (28.6)	19 (13.1)	12 (21.1)	43 (17.6)	0.051
BMI *, kg/m^2^	25.6 ± 2.8	26.2 ± 2.9	26.9 ± 3.8	26.2 ± 3.1	0.202
Serum urate, mg/dL	7.5 ± 1.8	7.9 ± 2.2	8.3 ± 1.7	7.9 ± 2.0	0.140
Serum creatinine, mg/dL	1.13 ± 0.60	1.03 ± 0.23	1.19 ± 0.22	1.08 ± 0.33	<0.001
eGFR, mL/min/1.73 m^2^	75.69 ± 21.20	81.30 ± 19.14	67.57 ± 15.48	77.12 ± 19.50	<0.001
Comorbidity, number (%)					
Hypertension	20 (47.6)	55 (37.9)	27 (47.4)	102 (41.8)	0.355
Diabetes mellitus	2 (4.8)	11 (7.6)	4 (7.0)	17 (7.0)	0.939
Ischemic heart disease	2 (4.8)	2 (1.4)	0 (0)	4 (1.6)	0.179
Valvular heart disease	1 (2.4)	2 (1.4)	1 (1.8)	4 (1.6)	0.803
Arrhythmia	1 (2.4)	2 (1.4)	1 (1.8)	4 (1.6)	0.803
Stroke	5 (11.9)	6 (4.1)	2 (3.5)	13 (5.3)	0.129
Epilepsy	0 (0)	1 (0.7)	0 (0)	1 (0.4)	1.000
Concomitant medication ^†^, number (%)					
ACE inhibitors	1 (2.7)	0 (0)	1 (2.0)	2 (1.0)	0.168
Angiotensin receptor blockers	6 (16.2)	20 (16.3)	9 (18.4)	35 (16.7)	0.966
Beta-blockers	4 (10.8)	6 (4.9)	3 (6.1)	13 (6.2)	0.410
Calcium channel blockers	4 (10.8)	17 (13.8)	8 (16.3)	29 (13.9)	0.788
Diuretics	2 (5.4)	10 (8.1)	6 (12.2)	18 (8.6)	0.545
Median ULT duration, months (IQR)	27.1 (15.2, 39.9)	22.3 (16.6, 35.1)	32.0 (22.5, 47.2)	25.3 (17.5, 36.9)	0.001

ACE, angiotensin-converting enzyme; BMI, body mass index; eGFR, estimated glomerular filtration rate; IQR, interquartile range; ULT, urate-lowering therapy. Note: Except where indicated otherwise, values are expressed as mean ± SD. * Data are available for 34 allopurinol users, 128 febuxostat users, and 40 benzbromarone users. ^†^ Data are available for 37 allopurinol users, 123 febuxostat users, and 49 benzbromarone users.

**Table 2 jcm-08-01067-t002:** Patient characteristics by achievement of target serum urate concentration.

Characteristics	Final Serum Urate ≥ 6 mg/dL (*n* = 53)	Final Serum Urate < 6 mg/dL (*n* = 191)	*p* Value
Male sex, number (%)	52 (98.1)	184 (96.3)	1.000
Age, years	46.3 ± 14.8	52.2 ± 13.9	0.008
BMI, kg/m^2^	26.7 ± 3.3	26.1 ± 3.0	0.342
Serum urate, mg/dL	8.2 ± 1.9	7.9 ± 2.0	0.270
Serum creatinine, mg/dL	1.12 ± 0.55	1.07 ± 0.23	0.341
eGFR, mL/min/1.73 m^2^	79.72 ± 21.80	76.40 ± 18.81	0.274
XOI users, number (%)	40 (75.5)	147 (77.0)	0.855
Median ULT duration, months (IQR)	22.5 (16.0, 32.8)	26.7 (18.2, 37.9)	0.055

XOI, xanthine oxidase inhibitor. Note: Except where indicated otherwise, values are expressed as mean ± SD.

**Table 3 jcm-08-01067-t003:** Mean changes in eGFR according to the baseline eGFR categories.

Category	Number (%)	ΔeGFR during ULT		*p* Value
		Final Serum Urate ≥ 6 mg/dL	Final Serum Urate < 6 mg/dL	
**Normal** (eGFR ≥90)	65 (26.6)	−4.46 ± 9.71	−1.36 ± 14.12	0.385
**Mild renal impairment** (eGFR 60–89)	130 (53.3)	9.31 ± 18.89	5.69 ± 11.60	0.366
**Moderate renal impairment** (eGFR 30–59)	47 (19.3)	−0.35 ± 3.87	5.33 ± 11.64	0.019
**Total**	244 (100.0)	2.68 ± 15.53	3.90 ± 12.53	0.554

Note: eGFR values are expressed in mL/min/1.73 m^2^ with the mean ± SD and the Δ symbol indicates the difference between the consecutive values. Two subjects with severely reduced baseline eGFR values (<30 mL/min/1.73 m^2^) were not tabulated.

**Table 4 jcm-08-01067-t004:** Logistic regression of baseline variables associated with improvement in renal function.

Variables	Univariable ModelOR (95% CI)	*p* Value	Multivariable ModelOR (95% CI)	*p* Value
Age: ≥65 years	0.64 (0.33, 1.24)	0.182	0.31 (0.13, 0.75)	0.009
BMI: ≥30 kg/m^2^	0.63 (0.24, 1.66)	0.348		
Hypertension	0.97 (0.58, 1.63)	0.905		
Diabetes mellitus	0.94 (0.35, 2.56)	0.901		
Baseline renal function (reference: normal)				
eGFR: 60–89 mL/min/1.73 m^2^	2.14 (1.17, 3.94)	0.014	2.39 (1.22, 4.68)	0.012
eGFR: 30–59 mL/min/1.73 m^2^	1.77 (0.82, 3.79)	0.144	2.44 (0.96, 6.21)	0.060

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
