# Peer review of "The Role of a “Treat-to-Target” Approach in the Long-Term Renal Outcomes of Patients with Gout"

_jcm, 2019, doi:10.3390/jcm8071067_

Round 1

Reviewer 1 Report

Thank you for the chance to review this paper. I am always keen to see routinely collected health care data used for research purposes. However, this use has to be appropriate, and in this case, I have some serious concerns regarding the methodology employed.

Major comments:

Overall, my concerns are about the aim of the study, which I think needs clarifying, and the approach taken to achieve that aim. I think the authors are attempting to replicate a trial design, but have not done this particularly rigorously. I would recommend that they consider taking a ‘target trial’ approach (e.g. https://www.ncbi.nlm.nih.gov/pmc/articles/PMC5550532/). This would then imply, I think, that the authors are trying to replicate an effectiveness trial, but they seem to talk a lot about efficacy, which I’m not sure you could ever study in routinely collected data.

Most of my other points relate in some way to this need for a clearer rationale, but I hope they are helpful.

Figure 1: I think the rationale for receiving 12 months of ULT to be included in the sample needs to be explained more clearly in the text. Is this about having time to achieve target SUA? I assume this is what the “non-persistent” group in the figure is referring to (n=270), but I think more explanation of this and how it may bias the sample from all gout patients is needed.

Table 3: I am concerned that the effects seen in Table 3 may in part be explained by the reasons people did or did not achieve target SUA levels (i.e. confounding) and I don’t see any adjustment for this.

Table 4 and related analysis: I think more detail is needed here. What is the dichotomous outcome used in the logistic regression? All I can see in the Methods describes the continuous measurement of eGFR. I have assumed that the purpose of this model is to better understand at ULT treatment initiation who will see an improvement in eGFR. If this is the case, then final SUA, development of AKI and development of urolithiasis should not be included in the model, as they happen later.

Minor points:

Statistical analysis (line 123): what do the authors mean by qualitative variables in this context? Are they in fact categorical?

Statistical analysis (lines 127-8): Variables cannot be parametric or non-parametric. These terms should refer to the statistical tests used. Instead the authors might refer to Normally or non-Normally distributed variables.

Statistical analysis (lines 132-133): Similarly to above, I think there are some problems with terminology. It is not a “likelihood ratio logistic regression”, but a likelihood ratio test was used to compare nested logistic regression models and decide on the relevant predictors. Related to this, I don’t think the term “eliminate insignificant predictors” is quite right either, but could be removed if the earlier part of the sentence was improved.

Results (lines 207): multivariate should read multivariable, as this is multiple independent variables in the model, not multiple dependent variables.

Discussion (lines 276-8): whilst the restriction to patients with persistent ULT treatment may give a clearer picture of the efficacy of ULT, I don’t think this reflects the real world, or indeed what would occur in a trial (it would, I think, be the equivalent of a per protocol analysis).  

Discussion (line 310): This is the first mention of missing data. How were missing data handled?

Author Response

Replies to reviewer 1

Thank you for the chance to review this paper. I am always keen to see routinely collected health care data used for research purposes. However, this use has to be appropriate, and in this case, I have some serious concerns regarding the methodology employed.

Major comments:

Overall, my concerns are about the aim of the study, which I think needs clarifying, and the approach taken to achieve that aim. I think the authors are attempting to replicate a trial design, but have not done this particularly rigorously. I would recommend that they consider taking a ‘target trial’ approach (e.g. https://www.ncbi.nlm.nih.gov/pmc/articles/PMC5550532/). This would then imply, I think, that the authors are trying to replicate an effectiveness trial, but they seem to talk a lot about efficacy, which I’m not sure you could ever study in routinely collected data.

Most of my other points relate in some way to this need for a clearer rationale, but I hope they are helpful.

- Thank you for pointing out importance on effectiveness in your comment. We absolutely agree with you and we do acknowledge that an effectiveness research in this area can help providers make clinical decisions the best. However, we decided our study not be one-sided on either completely pragmatic (effectiveness) or explanatory (efficacy), but instead lie on a continuum between these poles. We do understand that it tends toward more explanatory since it is primarily aimed to test hypotheses in such a setting as confounders can be minimized to the greatest degree as possible. Even though gout is relatively more manageable than other chronic diseases, patients with gout are notorious for their poor compliance. We understand that this study may be different from the setting of prospective clinical trials where joint efforts of fully informed practitioners and patients can secure a certain level of compliance. Given this circumstance, we had hoped to put some emphasis on efficacy by selecting an appropriate population of interest. Accordingly, we have revised the manuscript regarding clarification of the protocol and possible limitations related.

Figure 1: I think the rationale for receiving 12 months of ULT to be included in the sample needs to be explained more clearly in the text. Is this about having time to achieve target SUA? I assume this is what the “non-persistent” group in the figure is referring to (n=270), but I think more explanation of this and how it may bias the sample from all gout patients is needed.

- We add a little explanation here. Although there exist wide variations in follow-up duration across the publications, we considered that a 12-month period would be sufficient enough to address the long-term efficacy of ULT in gout. As some prioritize efficacy data from RCTs, regarding novel agents for gout, for example, randomized, double-blind phase 3 trials for regulatory approval assessed the urate-lowering efficacy of febuxostat and lesinurad for 12 months 1-3. However, we agree that the 12-month period may not be enough by some means and we added a commentary to the text. Next, compliance is usually a major issue for real-world management of gout as shown in a comprehensive cohort study in UK 4. To eliminate confounders as much as possible, we deemed essential excluding non-persistent group. Therefore, we revised the manuscript and explained about the non-persistent group and possible bias from the studied population on methods and discussion.

1.        Becker MA, Schumacher HR, Jr., Wortmann RL, et al. Febuxostat compared with allopurinol in patients with hyperuricemia and gout. N Engl J Med. 2005;353(23):2450-2461.

2.        Dalbeth N, Jones G, Terkeltaub R, et al. Lesinurad, a Selective Uric Acid Reabsorption Inhibitor, in Combination With Febuxostat in Patients With Tophaceous Gout: Findings of a Phase III Clinical Trial. Arthritis Rheumatol. 2017;69(9):1903-1913.

3.        Bardin T, Keenan RT, Khanna PP, et al. Lesinurad in combination with allopurinol: a randomised, double-blind, placebo-controlled study in patients with gout with inadequate response to standard of care (the multinational CLEAR 2 study). Ann Rheum Dis. 2017;76(5):811-820.

4.        Scheepers L, Burden AM, Arts ICW, et al. Medication adherence among gout patients initiated allopurinol: a retrospective cohort study in the Clinical Practice Research Datalink (CPRD). Rheumatology (Oxford). 2018;57(9):1641-1650.

- The description of the study period has been changed as follows.

"All subjects had at least one measurement of serum parameters within one month prior to and more than 12 months after the index date, in order to address the long-term efficacy of ULT." (lines 110-1)

- We discussed the “non-persistent” group and possible bias in detail.

"In this regard, this study was aimed to test hypotheses in such a setting as confounders can be minimized to the greatest degree as possible. In order to prevent yielding misleading outcomes on efficacy of ULT, we reviewed the persistence of screened patients before embarking on this study to exclude poorly compliant subjects. However, due to this strict enrollment, care should be taken when generalizing the results to patients in routine clinical practice." (lines 284-9)

Table 3: I am concerned that the effects seen in Table 3 may in part be explained by the reasons people did or did not achieve target SUA levels (i.e. confounding) and I don’t see any adjustment for this.

- We fully agree with the comment. We have presented characteristics of patients with moderate renal impairment at baseline.

"No significant difference was found between the two groups of demographic and clinical characteristics at baseline, with the use of angiotensin-converting enzyme inhibitors or angiotensin receptor blockers and diuretics."  (lines 194-7 & Supplementary Table S1)

- We added discussion of limitations in interpreting the results of this study.

"Since trajectories of change in serum urate and eGFR over the course of treatment were not incorporated in this study, the pattern of improvement in eGFR after ULT had been commenced cannot be detected." (lines 323-5)

"We acknowledge that the sample size was not large, with a minimum follow-up duration of 12 months, which might have resulted in a lack of statistical power to identify all aspects of the reno-protective effect attributable to ULT." (lines 330-2)

Table 4 and related analysis: I think more detail is needed here. What is the dichotomous outcome used in the logistic regression? All I can see in the Methods describes the continuous measurement of eGFR. I have assumed that the purpose of this model is to better understand at ULT treatment initiation who will see an improvement in eGFR. If this is the case, then final SUA, development of AKI and development of urolithiasis should not be included in the model, as they happen later.

- We gratefully appreciate your comment. We have modified the method and results of the logistic regression analysis according to your comment and revised the paragraph so that the study outcomes and variables of the logistic analysis are clearly presented.

"The primary objective of this study was to evaluate the impact of serum urate reduction below target concentration of 6 mg/dL on renal function as measured by changes in eGFR values. The sequential comparisons of eGFR from baseline to the last follow-up were computed for the categorized eGFR at baseline. The predictor variables (old age, obesity, comorbid hypertension or diabetes, and categorized eGFR) obtained at the baseline were assessed to find out its relationship to a net increase in eGFR during ULT using a logistic regression model." (lines 111-7 & Table 4)

Minor points:

Statistical analysis (line 123): what do the authors mean by qualitative variables in this context? Are they in fact categorical?

- The authors appreciate the reviewer’s comment. We changed ‘qualitative’ to ‘categorical’. (line 128)

Statistical analysis (lines 127-8): Variables cannot be parametric or non-parametric. These terms should refer to the statistical tests used. Instead the authors might refer to Normally or non-Normally distributed variables.

- The authors appreciate the reviewer’s comment. We changed ‘parametric’ to ‘normally distributed’ and ‘non-parametric’ to ‘non-normally distributed’. (lines 132-3)

Statistical analysis (lines 132-133): Similarly to above, I think there are some problems with terminology. It is not a “likelihood ratio logistic regression”, but a likelihood ratio test was used to compare nested logistic regression models and decide on the relevant predictors. Related to this, I don’t think the term “eliminate insignificant predictors” is quite right either, but could be removed if the earlier part of the sentence was improved.

- The authors appreciate the reviewer’s comment. The corresponding phrase have been modified as follows.

"A multivariable logistic regression analysis was performed to evaluate the factors associated with the improvement in renal function," (line 137-8)

Results (lines 207): multivariate should read multivariable, as this is multiple independent variables in the model, not multiple dependent variables.

- Thanks for the reviewer’s comment. It was changed accordingly. (line 221)

Discussion (lines 276-8): whilst the restriction to patients with persistent ULT treatment may give a clearer picture of the efficacy of ULT, I don’t think this reflects the real world, or indeed what would occur in a trial (it would, I think, be the equivalent of a per protocol analysis).

- The authors totally agree with the reviewer’s comment. We have revised the contents of the paragraph and now include a detailed explanation.

"In order to prevent yielding misleading outcomes on efficacy of ULT, we reviewed the persistence of screened patients before embarking on this study to exclude poorly compliant subjects. However, due to this strict enrollment, care should be taken when generalizing the results to patients in routine clinical practice." (line 285-90)

Discussion (line 310): This is the first mention of missing data. How were missing data handled?

- The data we have analyzed are missing only for BMI and the use of concomitant medications for few patients. We believe that the main outcomes would not be expected to significantly affect with some omission of the relevant part. (lines 322-3)

Reviewer 2 Report

In this article, authors presented their finding on urate lowering therapy in gout patients. Authors found treat-to-target approach is associated with better renal outcomes in individuals with renal impairment. Overall, this is a well-written and well-presented manuscript with positive data in managing renal function in gout patient. 

I will encourage authors to read the article of gout nomenclature recently published for the consensus of using correct terms when referring to gout and its related complications (Gout, Hyperuricemia and Crystal-Associated Disease Network (G-CAN) consensus statement regarding labels and definitions for disease elements in gout, PMID 29799677). For example, uric acid can be written as urate, and gout attack can be written as gout flares. 

Author Response

Replies to reviewer 2

In this article, authors presented their finding on urate lowering therapy in gout patients. Authors found treat-to-target approach is associated with better renal outcomes in individuals with renal impairment. Overall, this is a well-written and well-presented manuscript with positive data in managing renal function in gout patient. 

I will encourage authors to read the article of gout nomenclature recently published for the consensus of using correct terms when referring to gout and its related complications (Gout, Hyperuricemia and Crystal-Associated Disease Network (G-CAN) consensus statement regarding labels and definitions for disease elements in gout, PMID 29799677). For example, uric acid can be written as urate, and gout attack can be written as gout flares. 

- Thank you for your kind remarks on our study. We also appreciate your guide on nomenclature. They have been revised accordingly, including the consensus label “urate” and “gout flare”.

Reviewer 3 Report

Thank you for the opportunity to review this paper. My comments are as follows:

Abstract: Multivariate should be multivariable. Abstract is appropriate.

Introduction: Appropriate and clear.

Methods: line 86, why was <20 years used as the cut off as opposed to <18 years?

Line 133 - "insignificant" should be non-significant

Line 134 - should be "Results of the logistic..."

Results: line 146 should be "de novo ULT which continued.."

Line 171 to 177 is difficult to follow, are the numbers needed in the text?

Line 207 and Table 4 - multivariate should be multivariable.

Discussion: Appropriate.

Line 243 "A significantly lesser likelihood.." Should this be "statistically significantly lower likelihood.."

Tables and figures are appropriate.

Author Response

Replies to reviewer 3

Thank you for the opportunity to review this paper. My comments are as follows:

Abstract: Multivariate should be multivariable. Abstract is appropriate.

- Thanks for the reviewer’s comment. It was changed accordingly. (line 19)

Introduction: Appropriate and clear.

Methods: line 86, why was <20 years used as the cut off as opposed to <18 years?

Line 133 - "insignificant" should be non-significant

Line 134 - should be "Results of the logistic..."

- The authors appreciate the reviewer’s comment.

The legal standard of adults defined in the civil code of the Republic of Korea have been amended from the existing 20 years old to 19 years old in 2013.

The sentence has been modified, and it doesn’t contain the word “insignificant” now.

It has been changed accordingly. (line 139)

Results: line 146 should be "de novo ULT which continued."

Line 171 to 177 is difficult to follow, are the numbers needed in the text?

Line 207 and Table 4 - multivariate should be multivariable.

- The authors appreciate the reviewer’s comment.

‘Which’ has inserted appropriately. (line 151)

The following has been deleted.

"The mean eGFR was significantly increased (77.12 ± 19.50 mL/min/1.73 m2 vs. 80.76 ± 21.02 mL/min/1.73 m2, p < 0.001) throughout the study period. Improvement in renal function, as evidenced by an increase in eGFR from baseline, was reported in 147 (60.2%) patients."

They have been changed accordingly. (line 221 & Table 4)

Discussion: Appropriate.

Line 243 "A significantly lesser likelihood.." Should this be "statistically significantly lower likelihood."

- Thanks for the reviewer’s comment. It was changed accordingly. (line 252)

Tables and figures are appropriate.

Round 2

Reviewer 1 Report

Thank you for your responses to my original review. 

I appreciate the additional detail on Table 3 and the changed made in response to my original comments.

However, I think it is still unclear what the exact outcome is in the logistic regression presented in Table 4 and I continue to be a little confused by the rationale regarding assessment of efficacy in this real world data set.

Author Response

Replies to reviewer 1

Thank you for your responses to my original review.

I appreciate the additional detail on Table 3 and the changed made in response to my original comments.

However, I think it is still unclear what the exact outcome is in the logistic regression presented in Table 4 and I continue to be a little confused by the rationale regarding assessment of efficacy in this real world data set.

- Thanks for the reviewer’s comment. Improvement in renal function, as evidenced by an increase in eGFR from baseline, was reported in 147 (60.2%) patients during the study period. Thus, we tried to evaluate the factors associated with the improvement in renal function following ULT as a function of baseline demographic and clinical variables. The dependent variable used in logistic regression analysis as a dichotomous binary variable was an increase in eGFR, which means that the last measured eGFR value is greater than the baseline value. We expect that this result can help us to achieve an optimal reno-protective effect in addition to achieving the target serum urate concentration. The manuscript has been revised to address your comments. (Lines 115-7 and 220-3)

- We do understand that there is a continuum in the study features of traditional RCTs and real-world studies, with external validity increasing as more real-world features are included in the design (BMJ. 2018; 360: j5262). In light of the aim of this study, which is to build a body of evidence to support the emerging concept, the findings have been derived from relatively controlled circumstances under real-world conditions. We have considered that, and we have decided that it would be appropriate to modify the manuscript using "effectiveness" instead of "efficacy" in accordance with your suggestion. (line 111,  287 and 314)
